# Fatty Acid Profiles of Some Siberian Bryophytes and Prospects of Their Use in Chemotaxonomy

**DOI:** 10.3390/biom13050840

**Published:** 2023-05-15

**Authors:** Irina P. Filippova, Olesia N. Makhutova, Valeriya E. Guseynova, Michail I. Gladyshev

**Affiliations:** 1School of Fundamental Biology and Biotechnology, Siberian Federal University, 79 Svobodny av., Krasnoyarsk 660041, Russia; 2Institute of Biophysics, Krasnoyarsk Scientific Center, Siberian Branch, Russian Academy of Sciences, Krasnoyarsk 660036, Russia

**Keywords:** bryophyte, mosses, liverworts, fatty acids, chemotaxonomy

## Abstract

The composition of fatty acids (FAs) in gametophyte samples of 20 Siberian bryophyte species from four orders of mosses and four orders of liverworts collected in relatively cold months (April and/or October) was examined. FA profiles were obtained using gas chromatography. Thirty-seven FAs were found, from 12:0 to 26:0; they included mono-, polyunsaturated (PUFAs) and rare FAs, such as 22:5n–3 and two acetylenic FAs, 6a,9,12–18:3 and 6a,9,12,15–18:4 (dicranin). Acetylenic FAs were found in all examined species of the Bryales and Dicranales orders, dicranin being the predominant FA. The role of particular PUFAs in mosses and liverworts is discussed. Multivariate discriminant analysis (MDA) was performed to determine whether FAs can be used in the chemotaxonomy of bryophytes. Based on the MDA results, FA composition is related to the taxonomic status of species. Thus, several individual FAs were identified as chemotaxonomic markers at the level of bryophyte orders. These were 18:3n–3; 18:4n–3; 6a,9,12–18:3; 6a,9,12,15–18:4; 20:4n–3 and EPA in mosses and 16:3n–3; 16:2n–6; 18:2n–6; 18:3n–3 and EPA in liverworts. These findings indicate that further research into bryophyte FA profiles can shed light on phylogenetic relationships within this group of plants and the evolution of their metabolic pathways.

## 1. Introduction

Bryophytes are a separate evolutionary group within the lineage of higher plants (embryophytes). Their specificity is determined by the dominant role of the gametophyte, or sexual generation, in their life cycle. This results in their morphology: among these plants there are no tree-like or relatively large species. The largest linear size—up to 2 m—is characteristic of the aquatic moss *Fontinalis antipyretica*. Many bryophytes are so peculiar that it is a major challenge to reveal their taxonomic relationships.

The diversity of bryophytes on the earth is represented by 16,000–20,000 species within three divisions (Anthocerotophyta, Marchantiophyta and Bryophyta). The distinguishing features of the divisions include a sporophyte and gametophyte structure. In liverworts (the Marchantiophyta), gametophytes are dorsoventral with unicellular rhizoids and oil bodies in the cells, while green mosses (the Bryophyta) have gametophytes with or without multicellular rhizoids. The Marchantiophyta include thallus species with both simple and compound thalli and leafy species with obliquely inserted leaves. The Bryophyta species are leafy plants with leaves inserted at a right angle; they vary in peristome type and location of archegonia formation [1]. In liverworts and green mosses, the sporophyte consists of a foot, seta and capsule. The capsule of liverworts opens by splitting into valves; spores and elaters are formed inside. The protonema is usually reduced to a few cells [2]. The capsule of green mosses opens by a lid (the operculum); in the center of the capsule, there is a columella, a column of sterile tissue; the capsule opening is surrounded with a hygroscopic teeth-like peristome. Spores germinate to form filamentous protonemata [1].

Research into the chemical composition of bryophytes has shown that, among other features, they differ from vascular plants by their fatty acid (FA) profiles. They both contain FAs with 16 and 18 carbon atoms; as integral components of membrane lipids, they are universal for all plant species [3]. However, unlike vascular plants, bryophytes are able to synthesize long-chain polyunsaturated fatty acids (PUFAs), such as arachidonic (20:4n–6, ARA) and eicosapentaenoic (20:5n–3, EPA) acids [4,5,6,7,8,9,10]. Because bryophyte ancestors are green algae (e.g., [11,12,13]), incapable of ARA and EPA synthesis [14], they must have developed this ability anew.

PUFAs play an important role as building blocks of glycolipids and phospholipids, and metabolic precursors of oxylipins [15,16]; the latter function as highly active signaling molecules involved in the regulation of growth, development and senescence of living organisms, as well as in the protective responses of plant cells [11]. Their role in bryophytes may deserve special attention, as bryophytes resist being intensively consumed in food chains regardless of their nutritional value, comparable to that of monocotyledon plants [17]. Bryophyte protection may be provided by secondary metabolites, e.g., terpenes, as well as oxylipins that deter potential consumers. However, some highly specialized invertebrates may be connected trophically with individual taxa of bryophytes [18]. As a response to damage, up to half of all resources of linoleic, alpha-linolenic, arachidonic and eicosapentaenoic acids are used for oxylipin synthesis during the first hours [19]. This way, a relatively high content of PUFAs, in particular ARA and EPA, in some bryophyte species might enable them to face environmental impacts.

Among PUFAs, EPA is of major physiological importance for many animals, including humans (see the review [20]). Unlike ARA, EPA reduces the synthesis of prostaglandins, thromboxanes and leukotrienes in the human body, and reduces inflammation (e.g., [21]). In the plant kingdom, this n–3 PUFA is synthesized by some groups of marine and freshwater algae, except for green algae [14], while tracheophytes are not capable of EPA production and terrestrial biocenoses are deficient in it [20]. For this reason, exploring bryophytes as a source of this essential n–3 PUFA in terrestrial food webs is of special interest. As a way to enhance the nutritional value of angiosperm plants, attempts are being made to use bryophyte C20 PUFA synthesizing genes to obtain transgenic forms [9].

A fairly unusual metabolic group is acetylenic FAs that are only sometimes found as major components in certain seed oils of tracheophytes [22] but are present in lipids of mosses, fungi and algae. Acetylenic FAs have been found in a number of moss genera, e.g., *Fontinalis* [8], *Ceratodon* [23], *Rhodobrium* [24], *Anisothecium* [23], *Dicranella*, *Dicranum*, *Ditrichum* and *Dicranoweisia*, *Dicranodontium* [25,26], *Leptodontium*, *Syrrhopodon* and *Fissidens* [10], and some liverworts, namely, *Riccia* [27] (from trace amounts to ≈80%). The list of bryophyte species capable of acetylenic FA synthesis keeps being replenished with new species from different families and orders. These FAs may play various roles in plant metabolism. Dicranin is found both in the triacylglycerol (TAG) fraction [28] and in phospholipids; its biochemical precursor is 18:2n–6 [29,30]. Acetylenic FAs are used to store energy, especially when plant growth is retarded [28]. Dicranin is also supposed to be involved in protection against natural enemies. For example, after a phytophage (slug) attack, dicranin is used in the biosynthesis of cyclopentenone dicranenone A [31]. All of the above indicates that finds of rare and unique acetylenic FAs in bryophyte species can provide further insights into their biochemistry and physiology in general.

To identify bryophyte species that are difficult to distinguish, an integrative molecular and morphological approach is currently used (e.g., [32]). Over the past 20 years, the application of molecular methods has advanced the classification of bryophytes. Combining molecular studies with a thorough examination of intrinsic morphological features has enabled researchers to revise species affiliation to genera and families, to add new families and orders, or combine some orders and increase the number of large taxa [33,34,35,36]. For example, a new order, the Ptilidiales, was added; the Monocleales and Ricciales were assigned to the Marchantiales order; *Apomezgeria pubescens* was assigned to the *Mezgeria* genus [33]. The *Anomodon* genus was divided into four genera: *Anomodon, Pseudanomodon, Anomodontella* and *Anomodontopsis*. The *Heterocladium* genus was assigned to a new family and divided into two genera, *Heterocladium* and *Heterocladiella* [37]. Apparently, new advances in this field will result in further updates in the bryophyte taxonomy [33].

Recently, FA profiles have been extensively used to identify various groups of organisms, such as bacteria [14,38,39], sponges [40], microalgae [14,41,42] and fungi [43]. They have also become a promising tool for studying the evolution of biosynthetic pathways and clarifying phylogenetic relationships between taxa in the plant kingdom [44,45,46]. Taipale et al. [47] used FAs as chemotaxonomic markers to classify freshwater lake phytoplankton at the class level; in combination with the data on sterol composition, the researchers managed to identify phytoplankton genera. The composition of PUFAs was found to be specific for the divisions and classes of algae in marine phytoplankton [48]. Chemotaxonomic differences in the FA composition of seeds have been found between coniferous families [44]. Moreover, minor differences in the FA composition of seeds were not affected by edaphic or climatic conditions and allowed the clear distinction between a number of coniferous genera [49,50]. The question arises whether representatives of bryophyte taxa can also be distinguished using their FA profiles.

According to published research, FA content and composition varies considerably between different bryophyte species. In *Eurhynchium striatum*, the level of ARA was three times higher than that of EPA and amounted to 36% of total FA level, while in *Brachythecium rutabulum*, both ARA and EPA amounted to 23% [5]. In *Mnium hornum,* the level of ARA and EPA was 26% and 9%, respectively [6]. In contrast, in the studies by Dembitsky and Rezanka [7], only four moss species out of the thirteen examined, *Aulacomnium palustre*, *Bryum pseudotriquetum*, *Pseudobryum cinclidioides* and *Atrichum hassknechtii,* had significant ARA levels (4–6%), and the EPA level in them was 3 to 19 times lower; six other species contained approximately 3% EPA. The aquatic moss *Fontinalis antipyretica* contained up to 3.5% ARA and up to 6% EPA, depending on the season [8]. The thalli of *Marchantia polymorpha* contained 3% ARA and 6% EPA [9]. In their recent paper, Sarkar et al. [10] found 45 different FAs in 40 East-Himalayan moss species. In all the mosses, 16:0, 18:2n–6 and 18:2n–3 prevailed. Furthermore, certain representatives of the orders Bryales, Bartramiales and Hypnales differed from the other taxa by a high content of ARA. Statistical methods allowed the researchers to identify the content of ARA and the ratio of ARA/18:2n–6 as chemotaxonomic markers at the level of orders. However, studies on the chemotaxonomy of mosses and liverworts related to their FA composition are extremely scarce.

In this paper, we examine the FA composition of 20 most common Siberian bryophyte species from two divisions with the aim to identify possible chemotaxonomic differences. In addition, a likely role of bryophytes in food chains is discussed with respect to their FA profiles.

## 2. Materials and Methods

### 2.1. Sampling

Bryophyte samples were collected during the period 2019–2021, in natural biocenoses at three different sites: in the valley of the Laletin Stream, the right tributary of the Yenisei River (55°57′ N 92°44′ E); at the border of the Krasnoyarsk green belt (55°56′ N 92°44′ E); in the recreation zone of Krasnoyarsk Stolby National Park (55°54′ N 92°43′ E). To collect ontogenetically uniform specimens, samples were taken during the first week of April or the last week of October, relatively cold periods in the local climate, when all bryophytes have mature gametophytes and the growth is retarded. Samples included 20 bryophyte species from two divisions, four classes and eight orders most commonly found at the study site (Table 1).

Liverwort samples were collected from different clumps (there was a distance between clumps of more than 500 m). The Bryales species were sampled from different populations (at a distance of more than 4 km). The Dicranales species and *Thuidium assimile* were collected at a distance of up to 2 km between the sampling sites. The other species of the Hypnales and Polytrichales were sampled randomly from discrete, mainly single species clumps. Voucher specimens were deposited in the KRSU Herbarium.

Bryophyte samples were delivered to the laboratory in zipper bags, and on the same day cleaned of litter and other species under a binocular microscope. In all the species, there were no sporophytes on gametophytes. Parts of shoots or thalli (2–3 cm) were dried on filter paper, weighed on the analytical balance (±0.1 mg) and stored in Eppendorf tubes with a chloroform/methanol mixture (2:1, *v*/*v*) at –20 °C for further analysis.

### 2.2. Fatty Acid Analysis

The lipid extraction, lipid transesterification (methylation) and methyl esters purification methods used in this study are described in detail elsewhere [53]. The plant tissue was destroyed mechanically; lipids were extracted three times with a mixture of chloroform/methanol (2:1, *v*/*v*). Extracts were filtered through a layer of anhydrous Na_2_SO_4_; the solvent was removed using an RVO-64 rotary vacuum evaporator (Czech Republic). Dry lipids were supplemented with 1 mL of sodium methylate solution in methanol (8 g/L). The mixture was heated at 90 °C for 15 min, cooled, supplemented with 1.3 mL of methanol/H_2_SO_4_ mixture (97:3, *v*/*v*) and methylated at 90 °C for 10 min. FAMEs were extracted from the mixture with 2 mL of hexane and washed three times with 5 mL of saturated NaCl solution. Hexane was removed by rotary vacuum evaporation. FAMEs were resuspended in 0.1 to 0.3 mL of hexane prior to chromatographic analysis.

An analysis of fatty acid methyl esters (FAMEs) was conducted on a gas chromatograph with a mass spectrometer detector (Model 7000 QQQ, CA, USA) using a 30-m capillary HP-FFAP column (with an internal diameter of 0.25 mm). The conditions of the analysis were as follows: the velocity of the helium carrier gas was 1.2 mL/min; the temperature of the injection port was 250 °C; the temperature of the heater was increased from 120 to 180 °C at the rate of 5 °C/min (then kept isothermal for 10 min), then increased to 220 °C at the rate of 3 °C/min (then kept isothermal for 5 min) and finally increased to 230 °C at the rate of 10 °C/min (then kept isothermal for 20 min); the temperature of the chromatography/mass interface was 270 °C; the temperature of the ion source was 230 °C and that of the quadrupole was 180 °C; the ionization energy of the detector was 70 eV; scanning was performed in the range of 45–500 atomic units at the rate of 0.5 s/scan. Peaks of FAMEs were identified by their mass spectra, by comparing them to those in the integrated database NIST Mass Spectral Search 2.0 (build 22 October 2009) and to those in the standard mixture of 37 FAMEs (U-47885, Superco, USA).

For calculation of EPA content, mg/g of air-dry weight, an internal standard solution (a solution of 19:0 methyl ester in chloroform, 0.5 mg/mL; Sigma-Aldrich, St. Louis, MO, USA) was used. It was added to samples prior to lipid extraction.

To determine the position of triple bonds in acetylenic FAs, dimethyloxazoline (DMOX) derivatives of the FA fraction were obtained and their chromatography was performed similarly to FAMEs chromatography. A detailed description of obtaining DMOX derivatives is given in the paper by Kalacheva et al. [8]. Peaks of DMOX derivatives in the mass spectra were identified by comparing them to the database https://www.lipidmaps.org/resources/lipidweb/index.php?page=ms/dmox/dmox-acetylenic/index.htm (accessed on 12 October 2022). Examples of chromatograms of methyl esters of some FAs, mass spectra of two acetylenic acids, some polyunsaturated FAs, and one monounsaturated FA, and mass spectra of DMOX derivatives of two acetylenic FAs are presented in Appendix A.

### 2.3. Statistical Analysis

Mean and standard error calculations, normality tests, one-way analysis of variance (ANOVA) and multivariate discriminant analysis (MDA) were performed using STATISTICA software, version 9.0 (StatSoft, Inc., USA). Normality was tested by the Kolmogorov–Smirnov test. For normally distributed data, group comparisons were made using ANOVA with post hoc Tukey’s test. For non-normal distributions, the significance of differences was determined by the non-parametric Kruskal–Wallis test (KW). To improve the assumptions of normality and homogeneity of variance, the data on fatty acid composition (% of total FAs) used in the MDA were subjected to the arcsine square root transformation similarly to Kainz et al. [54] and Torres-Ruiz et al. [55].

## 3. Results

### 3.1. FA Composition

The results of the bryophyte FA composition analysis are given in Table 2. In the examined bryophyte species, 37 FAs were found, from 12:0 to 26:0. They included mono- and polyunsaturated (PUFAs); rare FAs, such as 22:5n–3 (Appendix A); two acetylenic FAs, 6a,9,12–18:3 and 6a,9,12,15–18:4 (dicranin) (Appendix A), and some saturated acids. Among them, 16:0 and 18:3n–3 were predominant in all the species. In the genus *Dicranum* (represented by three species, *D. fuscescen*, *D. polysetum* and *D. viride*), the average levels of 16:0, 16:1n–7 and 16:3n–3 were lower, and the level of 18:3n–6 was higher than in all the other species (Table 2, Appendix A). In *Dicranum* spp. and *Rhodobrium roseum*, the average levels of acetylenic FAs, 6a,9,12–18:3, 6a,9,12,15–18:4 and 18:4n–3, were similar; they were significantly higher than those in all the other bryophytes examined (Table 2). *Plagiomnium confertidens* differed from other bryophytes by higher levels of 18:0, 20:0, 22:0, 24:0 and 26:0. *Thuidium assimile* was characterised by higher levels of PUFAs, namely 20:3n–6, 20:4n–6, 22:4n–6 and 22:5n–3 (Table 2, Appendix A). Among the examined liverworts, the highest levels of 20:1n–9, 20:2n–6 and 25:1 were found in *Ptilidium ciliare* (Table 2). In *Marchantia polymorpha*, the levels of 14:0, 18:1n–7 and 20:5n–3 were, on average, higher, and the level of 20:4n–6 was significantly lower than in all other species. *Conocephalum conicum* had higher levels of 9:0, 16:0, 18:1n–9, 23:1 and 24:1 than the other species. In *Metzgeria pubescens*, the average levels of 15:0, tr16:1, 16:2n–6 and 18:2n–6 were higher, and the levels of 18:3n–6, 18:3n–3, 20:3n–6, 20:4n–3 and 20:5n–3 were lower than in the other species. In *Plagiochila porelloides*, 22:0 and 24:0 FAs were not found (Table 2).

There was only one replicate for some moss species, which did not allow us to assess the significance of differences in their FA levels in comparison with other species. However, we noted some distinctive features of their FA composition (Table 3). A rather high level of 16:3n–3 was found in *Polytrichum commune*, which is comparable to that in liverworts. The maximum level of 22:5n–3 was found in *Cratoneuron filicinum*; maximum levels of 20:0 and 22:0 were found in *Polytrichastrum longisetum* and 20:5n–3 in *Brachythecium rivulare* and *Entodon schleicheri*. The level of 18:4n–3 in *Rhytidium rugosum* and *Climacium dendroides* was high, similarly to *Rhodobrium roseum* and *Dicranum* spp. (Table 3, Appendix A).

For EPA, which can be considered as a marker of the trophic value of individual species, the content (mg/g of air-dry weight) was determined. The highest content of EPA was found in *Marchantia polymorpha, Rhytidium rugosum* and *Entodon schleicherii*, and the lowest in *Metzgeria pubescens, Polytrichastrum longisetum* and *Dicranum* spp. (Table 2 and Table 3). 

### 3.2. MDA Results

To identify phylogenetic differences in the FA composition of bryophyte species, an MDA was conducted, which allowed the pooling together of species belonging to the same order. Before MDA (forward stepwise procedure), to optimize variables vs. cases spreadsheet, acids with minor overall means (too many zero cases, 9:0, 20:1n–9, 23:1, 24:1, 25:1) were excluded one by one until the minimum tolerance appeared to be higher than the specified limit. Furthermore, during the analysis, FAs 16:1n–7 and 24:0 were automatically excluded from the model by the stepwise procedure because of their insignificant *F* values. The MDA demonstrated significant differences between the FA composition of bryophytes in different orders (Figure 1, Table 4). Root 1 discriminated best between the Bryales mosses and the Ptilidiales liverworts (Figure 1, Table 4). Variables that contributed the most to the first discriminant function (Root 1) were, on the one hand, 6a,9,12,15–18:4, 18:4n–3 and 6a,9,12–18:3, and on the other hand, 18:2n–6, 16:3n–3 and 16:2n–6 (Table 4). Root 2 discriminated best between the Jungermanniales and Ptilidiales liverworts and between the Hypnales and Dicranales green mosses (Figure 1, Table 4). The distinctions in Root 2 were primarily due to the contributions of 20:5n–3, 18:3n–3 and 20:4n–3 vs. 6a,9,12,15–18:4, 15:0 and 6a,9,12–18:3 (Table 4). The pattern of distribution of the orders from the Marchantiophyta and Bryophyta divisions observed in the resulting scatterplot also allowed distinction between these two divisions (Figure 1).

## 4. Discussion

The study showed that the FA composition of all examined bryophytes included a broad range of C16, C18, C20, C22 PUFAs, acetylenic FAs and also some monounsaturated and saturated FAs. It had some common features, e.g., 16:0 and 18:3n–3 prevailed in all the species. However, there was a great deal of variation in the FA levels between individual species and taxonomic groups. This is in agreement with the view that bryophyte FA composition is very special and requires a comprehensive examination [10].

Liverworts showed significantly higher levels of 16:0, 18:2n–6 than green mosses, which can be associated with the composition and thickness of their cuticles. Bryophyte gametophytes are covered with cuticular wax; according to recent research, they contain, among other compounds, FAs (16:0, 18:2n–6 [56,57]), with 16:0 being predominant in liverwort and green moss species [56]. Moreover, FAs are the major components in the cuticle of thallus liverworts (<80%) [56]. In liverworts, the cuticle was shown to be thicker than in green mosses [56], which can account for the observed differences between these two divisions in 16:0 and 18:2n–6 levels.

In the present study, special attention was given to EPA with respect to the potential trophic value of bryophytes. Some bryophytes in the surveyed region were shown to contain significant amounts of this essential long-chain PUFA. The maximum content of EPA was noted for *Rhytidium rugosum* (4.4 mg/g of air-dry weight) and *Entodon schleicherii* (4.0 mg/g) among mosses (the Hypnales), and for *Marchantia polymorpha* (5.4 mg/g) among liverworts (the Marchantiales). The findings of other authors [10,51] support our conclusion that the Hypnales species produce more EPA compared to other orders of Bryophyta. This biochemistry is consistent with the fact that the Hypnales occupies a terminal position in the moss phylogenetic tree [58,59]; many families of the Hypnales emerged less than 100 million years ago as a result of genome duplication under weaker purifying selection [60]; they then probably underwent a rapid diversification [59]. The appearance of the liverwort *Marchantia* is within the same time frame (21–70 Ma); *M. polymorpha* as a species is even younger (2–11 Ma) [61].

According to our data, the content of EPA in bryophytes is comparable to that in some algae, e.g., *Laminaria*, *Fucus* and *Phaeocystis* [20]. This allows us to hypothesize that bryophytes might be a source of EPA in terrestrial ecosystems, where vascular plants are not capable of its production. Further research on EPA consumption by primary consumers, and its transfer to other animals via food chains, could help to estimate the real ecological value of bryophytes. We can also speculate that the high content of EPA in bryophytes allows them to occur under a variety of environmental conditions and adapt to their change.

In some bryophytes, we also found C22 PUFAs, namely, 22:4n–6 and 22:5n–3. The maximum content of 22:4n–6 occurred in *Thuidium*, 22:5n–3 in *Cratoneuron filicinum*; furthermore, we found it in trace amounts in the majority of examined moss and liverwort species. Findings of 22:5n–3 in *Drepanocladus aduncus*, *D. tundrea*, *Leucobryum glaucum*, *Campylium*, *Cratoneuron* and *Bryum tortijdium* were reported earlier [62]. These mosses usually inhabit swamps, wet meadows or the banks of streams [59]. We sampled *Cratoneuron filicinum* at a confluence of a cold spring into the Laletina stream; it seems possible that 22:5n–3 might be involved in the mechanisms of survival at a low water temperature (the yearly maximum +10 °C).

An increase in n–3 PUFA synthesis was reported to accompany adaptation to low temperature conditions [63]. For example, a higher desaturase activity was determined in some East-Himalayan species of Hypnales and Bryales [10] and Marchantiales [9]. In *Marchantia* and a number of mosses, an increase in EPA and a decrease in ARA occurred after treatment with low temperatures [5,63,64]. To determine the extreme levels of PUFAs, we collected samples during the cold period. However, in our studies, the level of ARA varied greatly in both mosses and liverworts and was not related to taxonomic affiliation.

In this research, we found two acetylenic FAs, 6a,9,12–18:3 and 6a,9,12,15–18:4 (dicranin), in mosses of the Dicranales and Bryales orders. Dicranin dominated in FA profiles, both in *Rhodobrium roseum* (Bryales) and *Dicranum* spp. In *R. roseum*, dicranin content was similar to that in *R. ontariense* [24], but 6a,9,12–18:3 content was twice as low. FA profiles in three *Dicranum* species (*D. viride*, *D*. *polysetum* and *D*. *fuscescens*) were generally similar to those reported by Konh et al. [25] for the samples collected in Germany and Austria, except for acetylenic FAs. These researchers found dicranin in all three species, but 6a,9,12–18:3 in *D. viride* only. The level of dicranin also differed; in our study, dicranin amounted to 45 ± 6% (almost half) of the total FAs in all three *Dicranum* species, while in the German and Austrian samples the level of dicranin varied from 27% (*D*. *polysetum*) to 78% (*D*. *fuscescens*) [25]. In metabolism, dicranin is known to be a metabolic precursor of the oxylipins involved in anti-consumer defence [34]. In terms of chemotaxonomy, acetylenic FAs have been shown to lose their value as taxonomic markers of the Dicranales [10].

A comparison of our results with the reports by other authors on related species of the same genera showed that the level of some FAs was different, while the level of other FAs was almost the same. Thus, our data on the FA composition of *Plagiomnium confertidens* were consistent with those for the closely related species *P. cuspidatum* [65]. In our study, *Rhodobrium roseum* differed from *R. ontariense* [24] in the levels of some FAs, i.e., 16:0, 18:1n–9 and 6a,9,12–18:3; the level of them in the latter was twice as high. In our study, *Brachythecium rivulare* contained twice as much EPA, 18:1n–7, 18:1n–9, and less 16:1n–9 and 16:1n–7 than *B. salebrosum* examined by Hansen and Rossi [5]. Comparison of the FA composition of *Marchantia polymorpha* samples from Siberia with the data for samples obtained in Minnesota [66] showed that our plants contained twice as much 14:0, 16:1n–9 and 16:1n–7, but twice as little 16:2n–6 and ARA. In some cases, it was hardly possible to adequately compare our data with other reports on FA composition of the same species due to the difference in research methods or units of FA content measurement used.

The MDA demonstrated that the observed differences in FA profiles are not random but are related to the taxonomic status of the species. In the MDA, bryophyte species were grouped in orders according to their FA profiles. Moreover, the location of the orders in the scatterplot allowed distinction between these divisions, which correlates with the monophyletic origin of each of these divisions [58,67,68]. The discriminant variables which distinguished best between the liverworts and green mosses were 16:0 and 15:0 FAs, and 18:2n–6, 16:3n–3, 16:2n–6, 18:4n–3, 6a,9,12,15–18:4 and 6a,9,12–18:3 PUFAs. The latter two (acetylenic) acids were found in the Bryales and Dicranales. The highest levels of 16:0, 15:0 and 16:3n–3 were found in the Marchantiales, Metzgeriales and Jungermanniales. The Metzgeriales and Jungermanniales orders were also distinguished by the levels of 16:2n–6. Despite the low 18:4n–3 levels in the examined bryophyte species, the MDA showed that the differences were sufficient to distinguish between the orders.

The other PUFAs, 20:5n–3, 20:4n–3 and 18:3n–3, allowed discrimination between the examined orders of liverworts within the Marchantiophyta division and the orders of mosses within the Bryophyta division. These PUFAs were high in Hypnales and Jungermanniales. This is partially consistent with the data by other authors who noted that the share of C20 PUFAs in total lipid content varied in a number of moss species from different orders, e.g., high levels of ARA were observed in the Hypnales species [10]. However, in our study on comparing both mosses’ and liverworts’ total lipid FA profiles, we discovered that other polyenes, namely, C16 and C18 PUFAs, were meaningful for separating taxa, while ARA was not a significantly important taxonomic indicator. Similarly, several FAs (16:0, 18:0, 18:1n–9, 18:2n–9 and 18:3n–3) were identified as discriminants between coniferous families and genera using PCA analysis [44].

Some orders of green mosses and liverworts were also shown to differ in their polar lipid FA profiles [62]. Thus, levels of ARA and EPA in monogalactosyldiacylglycerols were the lowest in liverwort orders and the highest in Hypnales [62]. The total content of C20 PUFAs in phospholipids was three times lower in *Ceratodon purpureus* (the Dicranales), compared to *Pleurozium schreberi* (the Hypnales), while the content of C16 and C18 PUFAs was 1.5 times higher [69]. This allows one to hypothesise that the content and composition of polar lipids is an important part of the total lipid content of bryophyta and may correlate with the taxonomy of this group. However, biochemical transformation between the classes of polar lipids and triacylglycerols is possible [29] and can influence their ratio. For this reason, only tentative conclusions on a separate chemotaxonomic role of neutral and polar lipid FA profiles can be drawn.

Thus, the present analysis of the FA profiles of 20 species of Siberian mosses and liverworts showed that they contain a broad range of FAs which can perform a number of vitally important functions. Moreover, FA composition is related to the taxonomic status of species. The MDA results clearly indicate that several FAs can be potential chemotaxonomic markers at the level of bryophyte orders. For mosses, these are 18:3n–3, 18:4n–3, 6a,9,12–18:3, 6a,9,12,15–18:4, 20:4n–3 and EPA. For liverworts, these are 16:3n–3, 16:2n–6, 18:2n–6, 18:3n–3 and EPA. The relationship between the FA profiles of certain species and their taxonomic affiliation indicates that biochemical pathways for FA synthesis in bryophytes may vary depending on their taxa. To explore the observed patterns in bryophyte chemotaxonomy, further research on the identification of taxonomically significant FAs with involvement of a broader range of bryophyte species is required.

## Figures and Tables

**Figure 1 biomolecules-13-00840-f001:**
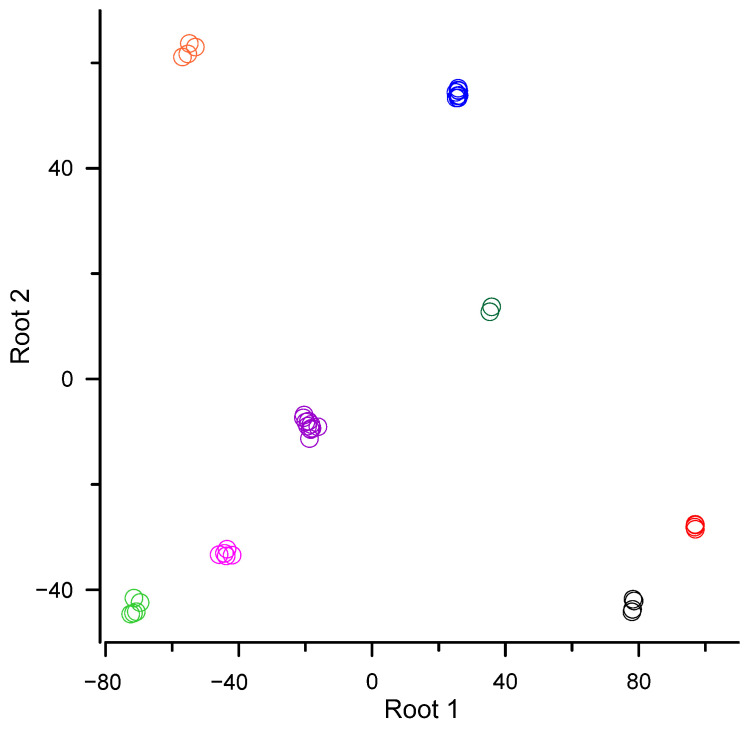
Scatterplot of canonical scores for two discriminant functions, Root 1 and Root 2, after the multivariate discriminant analysis (MDA) of the fatty acid (FA) composition (% of total FAs, arcsine square root transformation). Division Bryophyta: black circles—order Dicranales, red circles—order Bryales, dark green circles—order Polytrichales, blue circles—order Hypnales *; division Marchantiophyta: light green circles—order Ptilidiales, pink circles—order Metzgeriales, purple circles—order Marchantiales, orange circles—order Jungermanniales. * Based on 10 samples from 8 species (including 7 single-replicate samples).

**Table 1 biomolecules-13-00840-t001:** Bryophyte species examined: taxonomic affiliation and number of samples (*n*).

Division	Class	Order	Species	*n*
Marchantiophyta ^1^	Marchantiopsida	Marchantiales	*Marchantia polymorpha* L. (9001 KRSU) ^3^	7
			*Conocephalum conicum* (L.) Dumort. (9002 KRSU)	4
	Jungermanniopsida	Metzgeriales	*Metzgeria pubescens* (Schrank) Raddi (9011 KRSU)	5
		Ptilidiales	*Ptilidium ciliare* (L.) Hampe (9006 KRSU)	5
		Jungermanniales	*Plagiochila porelloides* (Torr. ex Nees) Lindenb (9009 KRSU)	4
Bryophyta ^2^	Polytrichopsida	Polytrichales	*Polytrichum commune* Hedw. (9046 KRSU)	1
			*Polytrihastrum longisetum* (Sw. ex Brid.) G.L.Sm. (9045 KRSU)	1
	Bryopsida	Dicranales	*Dicranum viride* (Sull. &Lesq.) Lindb. (9049 KRSU)	1
			*D. polysetum* Sw. (9030 KRSU)	2
			*D. fuscescens* Turner (9043 KRSU)	1
		Bryales	*Rhodobryium roseum* (Hedw.) Limpr. (9038 KRSU)	3
			*Plagiomnium confertidens* (Lindb. & Arnell) T.J.Kop. (9027 KRSU)	3
		Hypnales	*Anomodon attenuatus* (Hedw.) Huebener (9033 KRSU)	1
			*Brachythecium rivulare* Bruch et al. (9019 KRSU)	1
			*Entodon schleicheri* (Schimp.) Demet. (9050 KRSU)	1
			*Climacium dendroides* (Hedw.) F.Weber & D. Mohr (9020 KRSU)	1
			*Cratoneuron filicinum* (Hedw.) Spruce (9018 KRSU)	1
			*Neckera pennata* Hedw. (9040 KRSU)	1
			*Rhytidium rugosum* (Hedw.) Kindb. (9025 KRSU)	1
			*Thuidium assimile* (Mitt.) A.Jaeger (9015 KRSU)	3

^1^ Liverworts are classified according to Potemkin and Sofronova [51]. ^2^ Mosses are classified according to Ignatov et al. [52]. ^3^ Voucher specimens were deposited in the KRSU Herbarium.

**Table 2 biomolecules-13-00840-t002:** Mean levels of FAs (% of total ± standard error) and EPA content (mg/g of air-dry weight) in mosses. Means labelled with the same letters are not significantly different at *p* < 0.05 according to Tukey HSD post hoc test in the one-way analysis of variance (ANOVA). For non-normal distributions according to the Kolmogorov–Smirnov criterion (labelled with *), the significance of differences is confirmed by the non-parametric Kruskal–Wallis test (KW). Non-significant ANOVA or KW results are not labelled with letters or *.

Fatty Acid	*Dicranum* spp.	*Plagiomnium* *confertidens*	*Rhodobrium* *roseum*	*Thuidium* *assimile*	*Ptilidium* *ciliare*	*Conocephalum* *conicum*	*Metzgeria* *pubescens*	*Marchantia* *polymorpha*	*Plagiochila* *porelloides*
9:0 *	<0.1 ± <0.1	<0.1 ± <0.1	<0.1 ± <0.1	<0.1 ± <0.1	<0.1 ± <0.1	0.5 ± 0.1	<0.1 ± <0.1	<0.1 ± <0.1	<0.1 ± <0.1
12:0	<0.1 ± <0.1 ^A^	<0.1 ± <0.1 ^AB^	<0.1 ± <0.1 ^AB^	<0.1 ± <0.1 ^AB^	0.1 ± <0.1 ^AB^	0.2 ± <0.1 ^B^	0.1 ± <0.1 ^AB^	0.1 ± <0.1 ^AB^	<0.1 ± <0.1 ^AB^
14:0	0.2 ± 0.1 ^A^	0.4 ± 0.1 ^AB^	<0.1 ± <0.1 ^A^	0.3 ± 0.1 ^AB^	0.4 ± 0.1 ^AB^	0.9 ± 0.1 ^BC^	0.5 ± <0.1 ^AB^	1.1 ± 0.2 ^C^	0.3 ± <0.1 ^A^
15:0	0.2 ± 0.1 ^AB^	0.2 ± <0.1 ^AB^	<0.1 ± <0.1 ^A^	0.2 ± <0.1 ^AB^	0.5 ± <0.1 ^ABD^	0.5 ± <0.1 ^BD^	1.2 ± 0.1 ^C^	0.7 ± 0.1 ^D^	0.4 ± <0.1 ^ABD^
16:0 *	6.5 ± 0.5	14.7 ± 1.6	7.4 ± 1.9	13.0 ± 0.4	13.6 ± 0.6	29.0 ± 0.9	19.4 ± 0.2	22.5 ± 0.8	19.0 ± 0.2
16:1n–9	0.1 ± 0.0 ^A^	0.7 ± 0.1 ^AB^	0.2 ± <0.1 ^AC^	0.2 ± 0.1 ^AC^	0.5 ± <0.1 ^AB^	0.6 ± <0.1 ^AB^	0.3 ± <0.1 ^A^	0.9 ± 0.2 ^BC^	1.1 ± <0.1 ^B^
16:1n–7	0.1 ± <0.1 ^A^	0.8 ± 0.4 ^AB^	0.1 ± <0.1 ^AC^	0.4 ± 0.1 ^AD^	0.4 ± 0.1 ^AD^	0.9 ± 0.1 ^BD^	0.9 ± 0.1 ^BD^	1.3 ± 0.2 ^B^	0.5 ± <0.1 ^ACD^
tr16:1 *	0.1 ± <0.1	0.5 ± <0.1	0.1 ± 0.1	0.3 ± 0.1	0.3 ± <0.1	0.3 ± <0.1	1.0 ± <0.1	0.5 ± 0.1	0.4 ± <0.1
16:2n–6 *	0.2 ± 0.1	0.3 ± 0.1	0.4 ± 0.2	0.4 ± 0.1	0.4 ± <0.1	0.3 ± <0.1	4.2 ± 0.3	0.6 ± 0.1	2.4 ± 0.1
17:0	0.2 ± 0.1	0.3 ± 0.1	<0.1 ± <0.1	0.2 ± <0.1	0.2 ± <0.1	0.2 ± <0.1	0.2 ± <0.1	0.2 ± <0.1	0.2 ± <0.1
16:3n–3	0.3 ± 0.1 ^A^	0.8 ± 0.1 ^AB^	3.1 ± 0.7 ^ABC^	0.7 ± 0.3 ^AB^	3.7 ± 0.1 ^BC^	4.9 ± 0.2 ^CD^	6.8 ± 0.2 ^D^	6.6 ± 0.9 ^D^	4.8 ± 0.1 ^CD^
18:0 *	1.4 ± 0.6	4.5 ± 1.2	0.4 ± 0.1	0.9 ± 0.1	1.1 ± 0.1	4.1 ± 0.8	0.7 ± <0.1	1.5 ± 0.2	1.6 ± 0.2
18:1n–9	2.8 ± 0.4 ^AB^	5.1 ± 1.6 ^ABC^	0.7 ± 0.2 ^A^	2.5 ± 1.4 ^AB^	5.1 ± 0.4 ^BCD^	7.5 ± 0.3 ^C^	1.3 ± 0.1 ^A^	6.0 ± 1.0 ^BC^	2.3 ± 0.2 ^AD^
18:1n–7 *	0.6 ± 0.1	2.5 ± <0.1	0.8 ± 0.4	0.7 ± 0.3	1.1 ± <0.1	1.5 ± 0.1	1.3 ± 0.1	5.1 ± 0.8	2.8 ± 0.1
18:2n–6	8.6 ± 0.7 ^A^	7.1 ± 0.5 ^A^	4.4 ± 1.1 ^A^	16.4 ± 0.2 ^B^	29.1 ± 0.7 ^C^	8.3 ± 0.2 ^A^	38.5 ± 0.2 ^D^	10.3 ± 1.7 ^AB^	26.0 ± 0.2 ^C^
18:3n–6	1.9 ± 0.2 ^A^	1.6 ± <0.1 ^AB^	1.4 ± 0.1 ^ABC^	1.1 ± <0.1 ^BCD^	1.0 ± <0.1 ^CD^	1.2 ± <0.1 ^BC^	0.4 ± <0.1 ^E^	0.7 ± 0.1 ^DE^	0.6 ± <0.1 ^DE^
18:3n–3	14.4 ± 2.1 ^AC^	17.4 ± 0.5 ^AB^	16.3 ± 2.1 ^AB^	25.1 ± 1.1 ^B^	16.6 ± 0.4 ^AB^	14.6 ± 0.8 ^A^	7.5 ± 0.3 ^C^	21.2 ± 2.0 ^B^	20.3 ± 0.4 ^AB^
18:4n–3	1.2 ± 0.2 ^A^	0.3 ± <0.1 ^BCD^	1.7 ± 0.1 ^A^	0.3 ± <0.1 ^BD^	0.1 ± <0.1 ^BD^	0.5 ± <0.1 ^BC^	<0.1 ± <0.1 ^D^	0.8 ± 0.1 ^C^	<0.1 ± <0.1 ^D^
20:0	1.0 ± 0.5 ^AB^	1.4 ± <0.1 ^B^	0.1 ± <0.1 ^AC^	0.8 ± 0.1 ^ABC^	0.3 ± <0.1 ^AC^	0.7 ± <0.1 ^ABC^	0.3 ± <0.1 ^AC^	0.6 ± 0.1 ^ABC^	0.1 ± <0.1 ^C^
20:1n–9 *	<0.1 ± <0.1	0.1 ± 0.1	0.1 ± <0.1	<0.1 ± <0.1	0.5 ± <0.1	0.3 ± 0.1	<0.1 ± <0.1	<0.1 ± <0.1	<0.1 ± <0.1
20:2n–6 *	<0.1 ± <0.1	0.2 ± <0.1	0.2 ± 0.2	0.2 ± 0.1	1.8 ± 0.1	0.2 ± <0.1	0.1 ± <0.1	0.3 ± <0.1	0.2 ± <0.1
6a,9,12–18:3 *	3.4 ± 0.6	0.2 ± 0.2	6.9 ± 4.3	<0.1 ± <0.1	<0.1 ± <0.1	<0.1 ± <0.1	<0.1 ± <0.1	<0.1 ± <0.1	<0.1 ± <0.1
20:3n–6	0.4 ± 0.1 ^AC^	0.7 ± <0.1 ^AB^	0.9 ± 0.3 ^BD^	1.2 ± 0.4 ^B^	1.0 ± <0.1 ^B^	0.6 ± <0.1 ^ADC^	0.1 ± <0.1 ^E^	0.3 ± <0.1 ^CE^	0.2 ± <0.1 ^CE^
20:4n–6	8.7 ± 1.0 ^A^	25.1 ± 3.5 ^B^	8.3 ± 2.8 ^AD^	22.8 ± 2.1 ^B^	15.7 ± 0.3 ^C^	8.1 ± 0.6 ^A^	12.8 ± 0.2 ^CD^	3.7 ± 0.2 ^E^	8.3 ± <0.1
6a,9,12,15–18:4 *	45.5 ± 5.9	0.9 ± 0.9	42.4 ± 6.5	<0.1 ± <0.1	<0.1 ± <0.1	<0.1 ± <0.1	<0.1 ± <0.1	<0.1 ± <0.1	<0.1 ± <0.1
20:3n–3	<0.1 ± <0.1 ^A^	0.2 ± 0.2 ^ABC^	<0.1 ± <0.1 ^A^	0.4 ± 0.1 ^BC^	0.3 ± <0.1 ^BC^	0.2 ± <0.1 ^AB^	<0.1 ± <0.1 ^A^	0.4 ± 0.1 ^C^	0.2 ± <0.1 ^AB^
20:4n–3	0.2 ± 0.1 ^AC^	0.1 ± <0.1 ^ABC^	0.3 ± 0.2 ^AC^	0.2 ± 0.1 ^AC^	0.1 ± <0.1 ^AB^	0.2 ± <0.1 ^AC^	<0.1 ± <0.1 ^B^	0.3 ± <0.1 ^C^	0.3 ± <0.1 ^C^
20:5n–3	0.9 ± 0.1 ^AD^	7.4 ± 1.6 ^BC^	2.4 ± 0.5 ^ABD^	6.1 ± <0.1 ^ABC^	1.6 ± <0.1 ^AD^	8.0 ± 0.5 ^BC^	0.2 ± <0.1 ^D^	10.3 ± 1.5 ^C^	6.7 ± 0.2 ^BC^
22:0	0.3 ± 0.2 ^AC^	1.6 ± 0.2 ^B^	0.1 ± 0.1 ^AC^	1.1 ± 0.1 ^B^	1.4 ± 0.1 ^B^	1.3 ± <0.1 ^B^	0.1 ± <0.1 ^C^	0.5 ± 0.1 ^A^	<0.1 ± <0.1 ^C^
23:1 *	<0.1 ± <0.1	<0.1 ± <0.1	<0.1 ± <0.1	<0.1 ± <0.1	<0.1 ± <0.1	0.2 ± <0.1	<0.1 ± <0.1	0.1 ± <0.1	<0.1 ± <0.1
22:4n–6 *	<0.1 ± <0.1	0.1 ± <0.1	0.1 ± <0.1	0.7 ± <0.1	0.1 ± <0.1	<0.1 ± <0.1	<0.1 ± <0.1	<0.1 ± <0.1	<0.1 ± <0.1
22:5n–3 *	<0.1 ± <0.1	<0.1 ± <0.1	<0.1 ± <0.1	0.7 ± 0.1	<0.1 ± <0.1	<0.1 ± <0.1	<0.1 ± <0.1	<0.1 ± <0.1	<0.1 ± <0.1
24:0	0.4 ± 0.2 ^AD^	3.0 ± 0.8 ^B^	1.0 ± 0.5 ^ACD^	1.7 ± 0.2 ^AB^	1.7 ± 0.2 ^BC^	1.5 ± 0.1 ^AB^	0.5 ± <0.1 ^AD^	1.3 ± 0.3 ^AC^	<0.1 ± <0.1 ^D^
24:1 *	<0.1 ± <0.1	<0.1 ± <0.1	<0.1 ± <0.1	0.2 ± 0.1	<0.1 ± <0.1	1.3 ± 0.1	<0.1 ± <0.1	1.0 ± 0.1	0.1 ± <0.1
25:0 *	<0.1 ± <0.1	0.1 ± <0.1	<0.1 ± <0.1	0.1 ± <0.1	<0.1 ± <0.1	<0.1 ± <0.1	<0.1 ± <0.1	<0.1 ± <0.1	<0.1 ± <0.1
25:1 *	<0.1 ± <0.1	<0.1 ± <0.1	<0.1 ± <0.1	<0.1 ± <0.1	0.5 ± 0.1	<0.1 ± <0.1	0.3 ± <0.1	<0.1 ± <0.1	0.3 ± <0.1
26:0 *	<0.1 ± <0.1	1.4 ± 0.2	<0.1 ± <0.1	0.4 ± 0.1	0.5 ± 0.1	<0.1 ± <0.1	0.2 ± <0.1	0.1 ± 0.1	<0.1 ± <0.1
EPA	0.3 ± <0.1	1.2 ± 0.5	1.3 ± 0.1	0.9 ± 0.1	no data	3.8 ± 0.4	0.1 ± <0.1	5.4 ± 0.8	no data

**Table 3 biomolecules-13-00840-t003:** FA levels (% of the total) and EPA content (mg/g of air-dry weight) in single replicate samples of mosses.

Fatty Acid	*Anamodon* *attenuatus*	*Brachythecium* *rivulare*	*Entodon* *schleicherii*	*Climacium* *dendroides*	*Cratoneuron* *filicinum*	*Neckera* *pennata*	*Rhytidium* *rugosum*	*Polytrichum* *commune*	*Polytrihastrum* *longisetum*
9:0	0.0	0.0	0.0	0.0	0.0	0.0	0.0	0.0	0.0
12:0	0.0	0.0	0.0	0.0	0.0	0.0	0.0	0.0	0.0
14:0	0.2	0.2	0.0	0.2	0.1	0.3	0.1	1.6	0.3
15:0	0.2	0.3	0.2	0.1	0.2	0.2	0.1	0.2	0.1
16:0	11.5	14.6	12.8	8.1	12.3	9.2	7.6	12.5	11.5
16:1n–9	0.2	0.1	0.1	0.2	0.1	0.2	0.1	0.2	0.4
16:1n–7	0.3	0.4	0.2	0.1	0.4	0.5	0.2	0.3	0.4
tr16:1	0.3	0.7	0.3	0.6	0.7	0.6	0.2	0.2	0.1
16:2n–6	0.3	0.1	0.0	0.5	0.1	0.6	0.3	1.6	1.5
17:0	0.1	0.2	0.1	0.2	0.1	0.2	0.1	0.2	0.2
16:3n–3	0.5	0.3	0.1	1.2	1.7	1.7	1.0	4.4	2.9
18:0	0.5	1.0	0.7	1.2	1.7	0.7	0.4	1.1	1.9
18:1n–9	5.0	3.4	1.2	3.6	5.9	6.5	7.8	2.7	2.4
18:1n–7	1.2	1.2	0.7	0.6	1.1	2.2	1.3	1.1	1.8
18:2n–6	18.5	9.8	7.1	17.3	10.3	22.4	21.3	12.8	14.6
18:3n–6	0.9	0.9	0.7	3.9	1.0	2.5	2.0	0.8	0.8
18:3n–3	17.9	15.8	21.0	26.0	22.2	18.7	18.6	33.6	30.3
18:4n–3	0.2	0.2	0.2	1.5	0.8	0.7	1.6	0.1	0.1
20:0	0.1	0.1	0.2	1.1	0.2	0.6	0.2	1.4	3.2
20:1n–9	0.2	0.1	0.0	0.2	0.1	0.2	0.4	0.2	0.1
20:2n–6	0.6	0.1	0.2	0.4	0.1	0.4	1.0	0.3	0.3
6a,9,12–18:3	0.0	0.0	0.0	0.0	0.2	0.0	0.0	0.0	0.0
20:3n–6	1.6	0.9	1.7	0.7	0.9	1.3	3.4	1.1	0.6
20:4n–6	27.9	18.8	24.3	19.3	13.5	17.9	14.4	14.0	14.9
6a,9,12,15–18:4	0.0	0.0	0.0	0.0	2.1	0.0	0.0	0.0	0.0
20:3n–3	0.4	0.3	0.9	0.5	0.0	0.4	0.5	0.7	0.7
20:4n–3	0.2	0.3	0.6	0.2	0.5	0.3	1.2	0.2	0.1
20:5n–3	7.5	21.8	20.4	7.7	16.5	8.0	13.2	2.9	2.8
22:0	0.0	0.2	0.5	2.0	0.4	0.5	0.5	1.9	3.0
23:1	0.5	0.1	0.0	0.0	0.0	0.0	0.0	0.0	0.0
22:4n–6	0.0	0.0	0.0	0.1	0.0	0.2	0.1	0.2	0.2
22:5n–3	0.0	0.1	0.1	0.1	3.5	0.1	0.1	0.0	0.1
24:0	1.4	1.5	3.2	0.8	1.9	1.2	0.6	2.0	1.9
24:1	0.7	0.1	0.3	0.0	0.2	0.6	0.6	0.5	0.6
25:0	0.2	0.1	0.2	0.0	0.0	0.0	0.0	0.0	0.0
25:1	0.0	0.0	0.0	0.0	0.0	0.0	0.0	0.0	0.0
26:0	0.3	0.3	0.8	0.2	0.5	0.3	0.1	0.4	1.1
EPA	1.1	2.6	4.0	1.6	1.9	1.3	4.4	0.5	0.1

**Table 4 biomolecules-13-00840-t004:** Results of the MDA of the FA composition (% of total FAs) of mosses and liverworts of different orders.

	Root 1	Root 2
Canonical *R*	0.9998	0.9997
Chi−square	959	764
Degree of freedom	210	174
*p*	<0.0001	0.0001
Means of canonical variables:		
order Dicranales (Bryophyta)	78.2	−42.9
order Bryales (Bryophyta)	96.9	−28.0
order Hypnales (Bryophyta)	25.7	54.1
order Polytrichales (Bryophyta)	35.6	13.2
order Ptilidiales (Marchantiophyta)	−71.2	−43.4
order Marchantiales (Marchantiophyta)	−18.9	−8.8
order Metzgeriales (Marchantiophyta)	−43.9	−33.1
order Jungermanniales (Marchantiophyta)	−55.0	62.4
Factor structure coefficients:		
12:0	−0.009	−0.007
14:0	−0.011	−0.008
15:0	−0.031	−0.017
16:0	−0.027	<0.001
16:1n–9	−0.017	0.001
tr16:1	−0.014	0.002
16:2n–6	−0.032	<0.001
17:0	−0.002	−0.002
16:3n–3	−0.033	−0.014
18:0	0.001	−0.002
18:1n–9	−0.004	<0.001
18:1n–7	−0.007	0.003
18:2n–6	−0.036	−0.001
18:3n–6	0.015	0.001
18:3n–3	0.006	0.023
18:4n–3	0.022	−0.005
20:0	0.008	−0.004
20:2n–6	−0.026	−0.001
6a,9,12–18:3	0.021	−0.015
20:3n–6	0.011	0.015
20:4n–6	0.007	0.010
6a,9,12,15–18:4	0.027	−0.020
20:3n–3	−0.007	0.018
20:4n–3	0.009	0.022
20:5n–3	0.004	0.033
22:0	0.003	−0.008
22:4n–6	0.007	0.007
22:5n–3	0.003	0.013
25:0	0.005	0.007
26:0	0.003	−0.001

## Data Availability

Research data are readily available from the authors on request.

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
