# Peer review of "Fatty Acid Profiles of Some Siberian Bryophytes and Prospects of Their Use in Chemotaxonomy"

_biomolecules, 2023, doi:10.3390/biom13050840_

Round 1

Reviewer 1 Report

Manuscript ID- Biomolecules-2220816

Title: Fatty acid profiles of some Siberian bryophytes and prospects of their use in chemotaxonomy

Comments:

Authors have analyzed the fatty acid profiles of twenty Siberian bryophytes, and attempts were made to find out the significance of these fatty acids for taxonomic interpretation. The topic seems interesting, as bryophytes are mostly overlooked plant group, and their taxonomy is a huge area for study, many issues of which need to be resolved. However, I have a number of concerns, queries, and comments about the overall methodology and presentation of the manuscript.

·         Several fatty acids were separately considered as chemotaxonomic markers at the order level. To come to such an inference, analysis of only twenty bryophytes, that too from both liverworts and mosses, is not sufficient. How the conclusions can be made from analysis of a single genus from an order? Authors should add data of fatty acid profiles of more mosses from an order. The relevance of choosing the particular taxa out of a huge number of bryophytes is not clear. If there is any specific purpose of selecting the taxa, author should clearly mention that. If not, were they randomly chosen based on their availability?

·         Introduction part is poorly written, and it is way too elaborate with a number of information, not relevant to the theme of the study. Authors should extensively reframe the introduction by making it brief and to the point. Introductory part does not dwell on the taxonomic problems which can justify the relevance of the study. Authors should mainly focus on explaining important issues related to bryophyte taxonomy and why chemotaxonomy is required to solve the issues.

·         First two paragraphs (line 26-40) of introduction are all about general features of bryophytes. That much elaboration is not required. Authors can be more specific while mentioning comparative features of Bryophyta and Marchantiophyta.

·         Line 41-54- It is sufficient to only mention that the previous works of bryophytes were focused on the analysis of secondary metabolites. Information related to those metabolites and their functions are not relevant. If any of those metabolites were previously considered for taxonomic interpretation, authors should only mention that.

·         Line 73-103- These portions are not relevant to the theme of study. What is the need for the information related to oxylipins?

·         In Materials and method section, sampling process is ambiguous. Elaboration of sampling part is required, like how were the bryophytes collected, particularly in case of multiple numbers of samples of a species? A particular moss (of orders- Dicranales, Bryales) was collected from a single clump or different clumps of a population? If from different clumps, how far were the clumps from which the samples were collected? How were the epiphytic pleurocarpous mosses/ leafy liverworts collected? Were they collected from same plant or different plants? If so, how far were the collection zones? If the sampling was done from a single clump, then it should not be considered as separate samples.

·         It is not recommended to conclude anything based on chemical investigation of a single sample. Eleven out of twenty bryophytes were sampled once, and the data provided were from single replicate. Authors are suggested to do at least three sampling of a species for replicate analysis.

·         If the specimens were submitted to any herbarium, mention the voucher numbers and details of herbarium in Table 1.

·         Why relatively colder months were considered for collection of samples?

·         Line- 147- Elaborate how the mosses were dried, and then cleaned. How can proper cleaning of a dried sample be ensured? It is mentioned that the analysis was performed from gametophytes. So, it is expected that the sporophyte part was removed. This is not clearly written.

·         Line- 148- Were the samples powdered or the whole shoot or thalli were kept in solvent?

·         Line- 151- Authors should at least briefly mention the steps of lipid extraction, transesterification, purification with references. Total lipids were extracted for fatty acid analysis which includes both polar and storage lipids. Fatty acids of polar lipids are not usually stable parameter, as there can be changes in saturation level of fatty acids depending on environmental conditions. As fatty acids of polar lipids are not stable parameters, authors need to justify how they can be considered for chemotaxonomic study. Authors should focus on neutral lipids instead of total lipids.

·         Line- 153- Authors should briefly mention the temperature program and other conditions of GC-MS analysis.

·         Line- 158- Why only the EPA content (mg/g dw) was determined?

·         Line- 155- Which library was used to match the mass fragmentation data?

·         Line- 164- The reference Kalacheva et al. seems wrong.

·         Were all FAs considered as variables for MDA? If no, based on which parameter, some FAs were excluded for analysis?

·         In results, a separate section should be added first, giving a general note on overall FA composition and diversity. Authors are suggested to add the identification data (mass fragmentation pattern, m/z values with relative intensity etc.) of rare FAME & DMOX derivatives of FAs, especially of uncommon FAs such as 22:5n3, 20:4n3, 25:1 as supplementary material.

·         Table 2- Mention which Fas are aFA1 and aFA2.

·         Line- 210- Explain how can you conclude it is species specific.

·         Discussion part is also superficially written with irrelevant statements. Some portions are just observations that can be included in the result section. Justification of the result or the references justifying the results are missing. Some parts are repetitions from the introduction such as line-243. The total discussion part should be reframed by giving emphasis on justification of data with references related to their phylogenetic studies.

·         Line- 247- High content of 18:4n3 is contradicting with the result.

·         Line- 266- Not only ARA or EPA but also the relative content of C18 PUFAs vary significantly depending on environmental conditions. There are many references to this. Therefore, the claim is not valid.

·         Line- 306-308, line- 321-324- No relevance to the theme of the paper.

·         Line-325-334- Why significance of EPA is required in the discussion part?

·         Authors have claimed some FAs as chemotaxonomic markers for separate orders or divisions, but throughout the discussion section, no justification is given for the claim.  

The topic of the manuscript is interesting, and the study has a lot of scopes. Unfortunately, poor presentation, and unnecessary information which does not have any relevance to the theme of the study, have declined the standard of the manuscript. Thorough revision and reframing are necessary before reconsidering the manuscript for publication. I am suggesting the authors to consider more Siberian bryophytes from the same or other orders, and do at least three separate sampling of them to strengthen their manuscript.  

Reviewer 2 Report

The manuscript discussed the role of particular PUFAs in 20 species of Siberian mosses and liverworts using MDA, designating some FAs as chemotaxonomic markers of bryophytes. It is well organized and the data presented are clear and meaningful. However, the manuscript can be accepted after a few minor corrections.

Introduction

> line 66: In order to support the generality that bryophytes are distinguished from vascular plants by their ability to synthesize long-chain PUFAs, more evidence/references should be added.

>  line 88:  The sentence seems to be incomplete. 

> line 106: "in a number of mosses" -> should mention "genus", because the examples listed are the names of genus, not species.

 > line 116: Should rephrase the sentence. (cyclopentenone dicranenone A synthesis)

Materials and Methods

> This section mentions several times the expression "are described in detail elsewhere". The author should either give more detail about the methods or rephrase the sentence.

Discussion

> Typos should be edited: line 280 (oxilipins), line 323 (EFA)

> line 330: Period should be omitted before listing the examples

I suggest including the GC-MS spectrum in the supporting information.

Reviewer 3 Report

It is interesting study and findings. Fatty acids profile from 20 bryophyte (moss and liverworts) species, 2 divisions, 4 classes and 8 orders were analyzed and using MDA to find potential markers FA for taxonomic order.

While the scientific content is sound, it can be improved by adding some chromatographic figures, such as GC-MS chromatogram and spectra, especially for demonstrating the identification of acetylenic acid.

In the Materials and Methods section, the fatty acid analysis only show the references for FAME and DMOX derivatization. It needs to show what has been adapted in this study specifically (and supporting data as suggested above).

Some minor comments:

- In Table 2, 3, 4 there were the short name aFA1 and aFA2 which have not been introduced anywhere in the paper.

- in Table 3, the report of 0.00 +/- 0.00 is not statistically meaningful. If it is to be reported, the first significant digit can be recorded, such as 0.009 +/- 0.001, or reported as 'nd' ( not detected or lower than limit of detection).

- In Table 3, some of the standard errors are 0.0  (e.g. 0.5 +/- 0.0 for 15:0 from Conocephalum conicum) which is not statistically meaningful. The first significant digit of SE needs to be presented and the mean value is determined accordingly, such as 0.542 +/- 0.001.

In general, the study and results are interesting and I support the acceptance after minor revision.
